# Development and application of simulation modelling for orthopaedic elective resource planning in England

Alison Harper [1,2] Thomas Monks [1,2] Rebecca Wilson [3,4]
Maria Theresa Redaniel [3,4] Emily Eyles [3,4] Tim Jones [3,4]
Chris Penfold [5,6] Andrew Elliott,[7] Tim Keen [7] Martin Pitt [1,2]
Ashley Blom [8] Michael R Whitehouse [7] Andrew Judge [5,6]

¹University of Exeter Medical School, NIHR Applied Research Collaboration South West Peninsula, Exeter, UK
²University of Exeter Faculty of Health and Life Sciences, Exeter, UK
³NIHR Applied Research Collaboration West, Bristol, UK
⁴Population Health Sciences, University of Bristol, Bristol, UK
⁵NIHR Bristol Biomedical Research Centre, Bristol, UK
⁶University of Bristol, Bristol, UK
⁷North Bristol NHS Trust Southmead Hospital, Bristol, UK
⁸The University of Sheffield, Sheffield, UK

**Correspondence to**
Dr Alison Harper;
a.l.harper@exeter.ac.uk

## ABSTRACT

**Objectives** This study aimed to develop a simulation model to support orthopaedic elective capacity planning.

**Methods** An open-source, generalisable discrete-event simulation was developed, including a web-based application. The model used anonymised patient records between 2016 and 2019 of elective orthopaedic procedures from a National Health Service (NHS) Trust in England. In this paper, it is used to investigate scenarios including resourcing (beds and theatres) and productivity (lengths of stay, delayed discharges and theatre activity) to support planning for meeting new NHS targets aimed at reducing elective orthopaedic surgical backlogs in a proposed ring-fenced orthopaedic surgical facility. The simulation is interactive and intended for use by health service planners and clinicians.

**Results** A higher number of beds (65–70) than the proposed number (40 beds) will be required if lengths of stay and delayed discharge rates remain unchanged. Reducing lengths of stay in line with national benchmarks reduces bed utilisation to an estimated 60%, allowing for additional theatre activity such as weekend working. Further, reducing the proportion of patients with a delayed discharge by 75% reduces bed utilisation to below 40%, even with weekend working. A range of other scenarios can also be investigated directly by NHS planners using the interactive web app.

**Conclusions** The simulation model is intended to support capacity planning of orthopaedic elective services by identifying a balance of capacity across theatres and beds and predicting the impact of productivity measures on capacity requirements. It is applicable beyond the study site and can be adapted for other specialties.

## STRENGTHS AND LIMITATIONS OF THIS STUDY

⇒ The simulation model provides rapid quantitative estimates to support post-COVID-19 elective services recovery towards medium-term elective targets.
⇒ Parameter combinations include changes to both resourcing and productivity.
⇒ The interactive web app enables meaningful and relevant parameter changes by healthcare managers, planners and clinicians.
⇒ Patient attributes such as complexity are not included in the model but are reflected in variables such as length of stay and delayed discharge rates.
⇒ Theatre schedules are simplified, incorporating the five key orthopaedic elective surgical procedures.

## INTRODUCTION

Elective joint replacement comprises one of the highest volumes of elective procedures worldwide.[1] In the UK, orthopaedics has been the specialty under the most pressure in terms of performance against National Health Service (NHS) elective operational standards.[2] Prior to the COVID-19 pandemic, increased waiting times for elective orthopaedic surgery reflected limited NHS resources and the competing demands of rising emergency admissions that affect mixed sites. This has been particularly problematic during the winter months when emergency demand for hospital care is high.[3]

Hip and knee replacement are strongly cost-effective from both a societal and a health system perspective compared with non-surgical treatment.[4] Procedures that are not delayed are more cost-effective than delayed intervention, while patients delayed for surgery for more than 180 days have been shown to be at higher risk of poor outcomes.[4–6] Additionally, reviews have found that low surgical volume is associated with longer procedure times and lengths of stay and poorer patient outcomes including risk of revision.[7–9] Given substantial evidence that surgeons undertaking low volumes of specific surgical activities may increase costs and result in less favourable outcomes for patients, Getting it Right First Time (GIRFT)[10] published a set of recommendations, such as ring-fencing beds and improving criteria-led discharge, aimed at reducing the significant variation found in practice. While successful,[11] performance against core national standards

has continued to deteriorate, attributable to increasing demand and lack of available capacity.[2 12]

The effect of the COVID-19 pandemic on elective orthopaedic services has been to compound ongoing challenges, with larger waiting lists and a steep decline in performance. Following the pandemic, a deterioration has been found in the health of patients who have had elective joint replacement postponed.[13] In response to this situation, the government has committed 2-year revenue allocations to support integrated care boards (ICBs) to expand capacity. ICBs and primary and secondary care providers are required to develop plans to meet national objectives and local priorities, in particular, to eliminate elective waits of over 65 weeks by March 2024.[14] While central capital funding will be key to achieving this, maximising use of resources and reducing lengths of stay and delayed discharges are required to make effective use of available capacity, and are associated with improvements in patient care.[12]

ICBs are, therefore, currently working to secure capital funds by delivering business cases that evidence optimal capacity and productivity configurations considering activity, workforce, capital requirements and potential revenue. Simulation modelling can be used as a planning tool to provide supporting evidence by modelling various configurations of bed numbers, theatre capacity and ward stays to estimate resultant surgical throughput. In this paper, we describe the development and implementation of an interactive, free and open-source simulation model to support planning of ring-fenced elective orthopaedic capacity. The model is codesigned with North Bristol NHS Trust (NBT), and is designed to be reusable, generalisable and to provide rapid information for clinicians, business and service managers across a range of scenarios relevant to new orthopaedic capacity planning.

## METHODS

We developed a discrete-event simulation (DES) model (programmed in Python V.3.8) of surgical activity and ward stay in a proposed ring-fenced orthopaedic facility. DES allows processes and pathways to be modelled at the individual patient level, and to explore the potential impact of changes to the system without the costs and risks associated with real-world changes. It has been used for patient flow management, resource allocation and scheduling, for example, in sexual health,[15] stroke pathways[16] and orthopaedics.[17 18] The DES was developed to have the flexibility to answer a range of 'what-if' questions of interest to NBT, and is generalisable to other NHS Trusts for orthopaedic elective planning.

The DES model is free and open source, and is available as a web app: https://hospital-efficiency-project.streamlit. app/. To preserve code, we have permanently archived it using Zenodo (HEP | Zenodo).[19] All code has an MIT license allowing free reuse and adaptation by researchers, industry and the NHS. Our app provides a user-friendly, interactive interface for the DES, including instructions

for use and documentation, allowing NHS staff to experiment with model parameters and generate immediate results without the need to download and instal software. The model can also be adapted to other specialties.

The model is documented using Strengthening the Reporting of Empirical Simulation Studies (STRESS) reporting guidelines,[20] available in online supplemental material 1.

### Data and setting

NBT serves a population of approximately 1 million people, with an age profile in line with England. Routinely collected data from the NHS Trust was used to identify patients receiving elective joint replacement between January 2016 and December 2019.

The Trust's electronic health records (EHRs) were used to identify elective joint replacements using a combination of OPCS4 procedure and surgical site codes (online supplemental material 2). Five core elective orthopaedic surgical procedure types were identified and verified. A small number of short day-case 'hip resurfacing' surgeries (n=52) were removed from the dataset as they rarely use bed capacity. The five remaining surgical types were classified into two classes: (1) Primary (87%): (primary hip replacement (p-THR n=3057; 51%), primary knee replacement (p-TKR; n=2302; 38%), unicompartmental knee replacement (p-UKR; n=679; 11%)); (2) Revision (13%): (revision hip replacement (r-THR; n=482; 55%), revision knee replacement (r-TKR; n=392; 45%)). Most patients did not remain in hospital once they were medically fit for discharge, however a proportion of patients in the EHR had a recorded medically fit for discharge date which preceded their actual discharge date (n=529; 7.6%).

The DES requires parameters describing patient lengths of stay, hence statistical probability distributions were fitted to each category of surgical procedure using the EHR data. The length of stay parameters (procedure, mean days (u), SD days (SD)) are: p-THR, u=4.4, SD=2.9; p-TKR, u=4.7, SD=2.8; p-UKR, u=2.9, SD=2.1; r-THR, u=6.9, SD=7.0; r-TKR, u=7.2, SD=7.6; delayed discharge, u=16.5; SD=15.1), which are converted to lognormal parameters within the model. Lognormal distributions were used for sampling lengths of stay in all cases. The mean lengths of stay are high against national benchmarks, and a key focus of future activity is to reduce lengths of stay.[11 21]

### Orthopaedic surgical pathway

The DES model is a simplified, high-level representation of the system of interest, which simulates individual patient flow through the system over time. Our model assumes an infinite waiting list. Baseline surgical theatre scheduling rules define how patients enter the simulation model according to their surgical class (primary or revision). Baseline rules are as follows:

► Three theatre sessions per day, 5 days per week with no weekend activity.

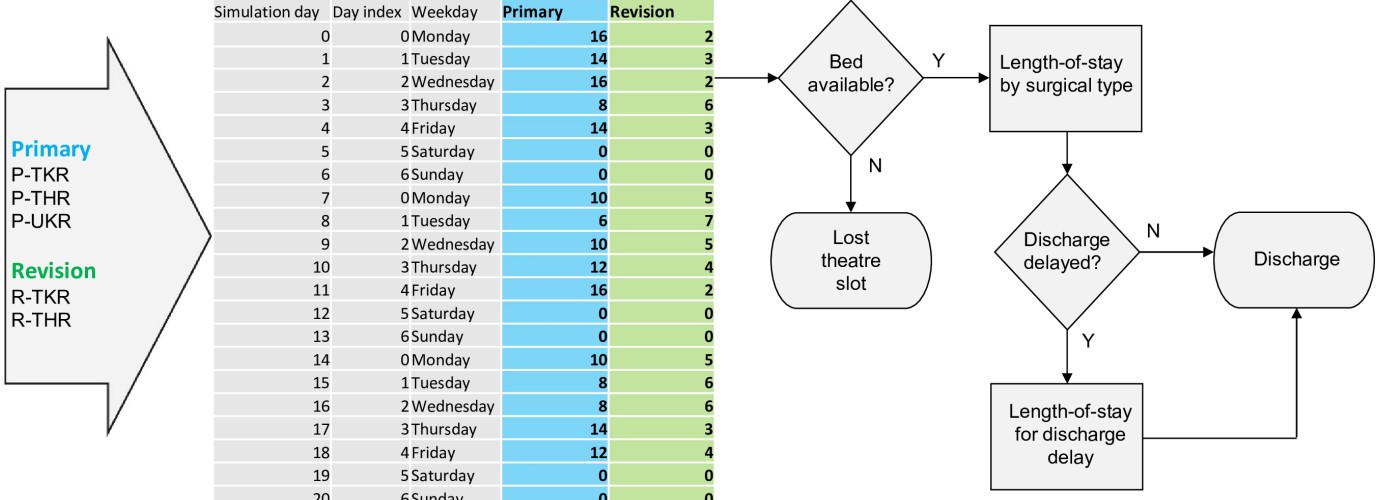

**Figure 1** Conceptual organisation of ring-fenced orthopaedic activity and ward stay. P-TKR, primary knee replacement; P-THR, primary hip replacement; P-UKR, unicompartmental knee replacement; R-TKR, revision knee replacement; R-THR, revision hip replacement.

▸ Morning and afternoon sessions schedule either: one revision or two primary surgeries.

▸ Evening session schedules one primary surgery.

Baseline resources for the model are 40 beds and 4 theatres. The DES samples a length of stay for each simulated patient from a lognormal distribution with parameters calculated from the previously fitted distributions by procedure type. Using stochastic routing, a length of stay is sampled for delayed patients. The simulation can be run using baseline parameters, and additional scenarios can be run by changing these parameters to determine the effects on model outputs, primarily surgical throughput. Figure 1 shows the organisation of surgical activity and ward stay.

## Model objectives

With the overall objective of maximising surgical throughput, the effects of changing model inputs can be investigated alone or in combination as experimental scenarios. These include:

▸ The number of theatres.

▸ The number of ring-fenced beds.

▸ Patient lengths of stay, including delayed discharges.

▸ Proportion of patients with a delayed discharge.

▸ Effects of running evening or weekend theatre sessions.

▸ Changes to surgical scheduling, for example, scheduling revision surgery (with longer, more variable lengths of stay) earlier in the week.

These questions are all in line with GIRFT priorities, which include accepting day surgery as default (where a bed is available as back-up), improving theatre utilisation and best practice care, and focused enhanced recovery. Higher surgical volumes, dedicated theatre teams and enhanced postoperative recovery are expected to improve patient outcomes. In turn, it is expected that lengths of stay will reduce, and standardised clinical pathways and discharge planning are likely to reduce discharge delays.[12 21]

The model outputs:

▸ Total surgical throughput: The primary goal of central capital funding allocations is to efficiently maximise surgical throughput,[14] so the configuration which best achieves this—within other constraints relevant to service planners such as workforce availability[22]—is a key output.

▸ Bed utilisation per day of week: For each experimental scenario, mean bed utilisation (occupancy) is outputted daily over model runtime. While there is no ideal average bed utilisation figure (which is dependent on many factors), it is commonly accepted that mean occupancies greater than 85%–90% can expect regular bed crises.[23] The results of excessive bed utilisation in the model can be seen as 'lost slots', where no bed is available for a patient scheduled for surgery. An additional consideration is that GIRFT recommend the extension of therapy services to support patient mobility goals towards discharge on any day with elective operating.[21] Therefore, theatre scheduling decisions (eg, 5–7 days service) are dependent on the availability of weekend staff.

▸ Lost theatre slots for system reasons: While beds are protected from outlying emergency admissions in a ring-fenced scenario, the balance of beds to theatre activity is a critical question. In the model, where patients are scheduled to arrive for surgery but no bed is available, the theatre slot is lost. In reality, other system behaviours will account for some of these lost slots. For example, the slot may be lost for patient reasons such as illness (ie, the patient does not attend for surgery or is deemed not fit for surgery at the point of admission); bed management activities may free up beds; or patients may be transferred to

**Table 1** Summary of scenarios varying procedure lengths-of-stay (los), bed numbers, proportion of patients with a delayed discharge (prop) and daily theatre schedule

| Scenario | los: 2 parameters | prop: 2 parameters | Beds: 9 parameters | Schedule: 2 parameters |
|---|---|---|---|---|
| Scenario 1 | Baseline | Baseline | 30–70 (in intervals of 5) | Baseline<br>Baseline+weekend |
| Scenario 2 | 0.25×baseline | 0.25×baseline | 30–70 (in intervals of 5) | Baseline<br>Baseline+weekend |
| Scenario 3 | Baseline | 0.25×baseline | 30–70 (in intervals of 5) | Baseline<br>Baseline+weekend |
| Scenario 4 | 0.25×baseline | Baseline | 30–70 (in intervals of 5) | Baseline<br>Baseline+weekend |

the acute hospital. In the model, lost slots (per day of week) are an indication of a mismatch between demand (theatre scheduling) and capacity (bed utilisation). In NBT, an average of 4.75 slots are lost per week for patient reasons, and a further 2.5 for system reasons.

### Patient and public involvement

Patients and the public were consulted in a workshop for suggestions and comments to inform the development of the grant that supported this work, and a further workshop informed scenarios used in model development.

### Role of the funding source

The funders had no role in any of the following: the study design; the collection, analysis and interpretation of data; the writing of the report; the decision to submit the paper for publication.

### RESULTS

A set of 72 indicative experimental parameter configurations were investigated (summarised in table 1) and described here:

Bed numbers between 30 and 70 beds (in increments of 5 beds; a total of 9 bed parameters) were investigated with each of 2 theatre schedules. The baseline theatre schedule is as described in the Methods section, with a 5-day working. The baseline+weekend schedule uses the same daily theatre allocations and theatre numbers, with a 7-day working. For each of these parameter changes, four scenarios were run for lengths of stay (baseline, 25% baseline) and for proportion delayed (baseline, 25% baseline). This totals 72 combinations.

The baseline procedure lengths of stay and proportion of patients with a delayed discharge are historical values described in the Methods section. As a goal of capacity planning is to reduce lengths of stay and delayed discharge, these are considered maximum values (high_los, prop_high), with minimum values set at 25% of baseline (low_los, prop_low). In all cases, the length-of-stay parameters for those patients with a discharge delay remain at the baseline value.

The results are plotted in figure 2. The top row displays the mean total daily surgical throughput for each scenario,

and for each theatre schedule. The middle row is mean daily bed utilisation. The bottom row displays 'lost slots', estimating the extent of the mismatch between patients scheduled and beds available.

The results show that the system is more sensitive to changes in procedure lengths of stay than to changes in the proportion of patients delayed, despite the long mean lengths of stay for delayed patients. At current procedure lengths of stay (high_los), bed utilisation is high with both current and reduced delayed discharges (scenarios 1 and 3). With reduced lengths of stay, the effects of reducing delayed discharges on required bed numbers are more significant, substantially reducing bed utilisation.

In the case of no weekend activity, a higher number of beds (65–70) than the proposed value (40 beds) will be required if lengths of stay remain unchanged. At this level of bed utilisation, reducing delayed discharges has little impact on required bed numbers. However, reducing lengths of stay in line with national benchmarks has enough impact on bed utilisation to allow for additional theatre activity.

Where lengths of stay can be reduced, weekend operating theatre activity (remaining at four theatres) increases surgical throughput, and beds remain underutilised in all cases above 40 beds. Reducing the proportion of patients with delayed discharge further reduces bed requirements to approximately 30 beds. Where lengths of stay remain at baseline values, weekend theatre activity cannot be considered, as bed utilisation and resultant lost slots are unacceptably high, even up to 70 beds. Users can investigate scenarios between these extreme values to gain realistic expectations of required bed and theatre numbers. Within the web application, results for bed utilisation include variability in output using boxplots. Sensitivity analysis and testing of parameters has been undertaken and is available to view in GitHub, Zenodo[19] and our online notebook (see the Data availability statement section).

### DISCUSSION

Our generalisable, open-access application allows those involved in planning the development and utilisation of ring-fenced elective orthopaedic units to rapidly model

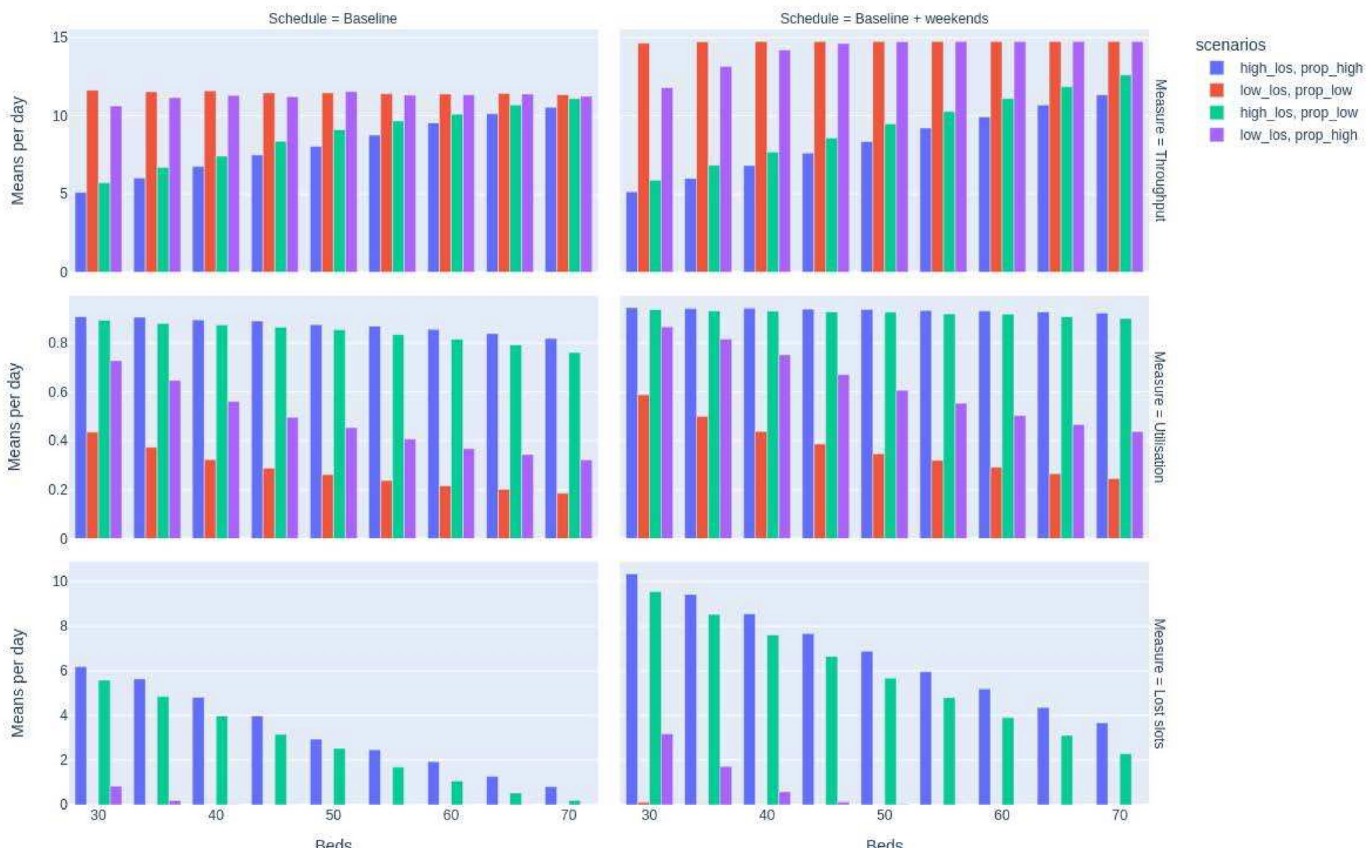

**Figure 2** Results of simulations across 30–70 beds, for each of 2 theatre schedules with 4 combinations of lengths of stay (baseline:high_los; baseline×0.25: low_los) and proportion delayed (baseline:prop_high; baseline×0.25: prop_los) for mean daily total surgical throughput, bed utilisation and lost slots.

different scenarios to predict delivery of elective surgical care. The model is adaptable to use local data for other units to model likely scenarios when planning activity. With minimal adaptation, it can also be applied to the delivery of other types of elective surgery.

Our experimental scenarios found that reducing lengths of stay in line with national benchmarks can increase surgical throughput and allow for additional theatre activity given the bed and theatre numbers initially proposed. We investigated weekend working, but the model also allows estimates to be obtained for increased theatre numbers, increased theatre utilisation (more procedures per day), and changes to theatre activity (eg, scheduling more complex surgery early in the week). However, if lengths of stay remain unchanged, proposed bed numbers in this instance will be inadequate. As procedure lengths of stay reduce, the effects of reducing the number of patients with a delayed discharge also become more significant. Reducing the length of the delay will similarly reduce bed utilisation and increase throughput, and the results of this can also be investigated using the simulation model.

A strength of simulation modelling is the use of underlying stochastic distributions using real-world data, as planning by average occupancy will not provide adequate reserve capacity to manage the variation seen in patient lengths of stay.[24] Our open approach to modelling is a further strength, as a range of scenario combinations can be investigated by users to support planning using the web app. Additionally, the model is available for reuse either through reparameterisation or adaptation. There are limitations to the use of simulation modelling. Assumptions and simplifications are required to convert a real-world problem into a computer representation. We assume that all historical (prepandemic) lengths-of-stay distributions fit current lengths of stay for baseline modelling. We do not account for patient frailty or other patient attributes beyond length of stay and the proportion of delayed discharges, and patients who have been delayed may be more complex. Additionally, our theatre scheduling rules include only the five main elective orthopaedic procedures. While other procedures may use theatre activity, they are not accounted for in bed planning, although simpler procedures will more likely be performed as day cases, and more complex procedures (such as spinal surgery) will be performed in a main hospital setting with high-dependency facilities.

Previous modelling and simulation work has focused on resource sharing for elective and non-elective joint replacement[17] and detailed studies of individual orthopaedic services.[18 25] Although the need for reusable models for orthopaedic wait list planning is recognised,[26]

this is the first free and open-source, generalisable DES model to support resourcing and efficiency of elective services that is available for use in this area.[27] Our model provides quantitative outputs estimating the effects on surgical throughput (per procedure); daily bed utilisation of changes to bed and theatre capacity, theatre scheduling, patient lengths-of-stay and discharge delays; and lost theatre slots, representing system pressure on beds. The model is designed for use by health services planners and clinicians, and is available as a free web-app to address usability and accessibility of results.[28] It is being used to evidence service configurations for the business case in NBT and is more widely applicable.

## CONCLUSIONS

Postpandemic, pressure to restore elective surgeries against new interim national targets necessitates efficient and effective use of allocated public funds. This planning is happening rapidly and on a large scale across England. Simulation modelling offers an effective method for planning elective services, identifying a balance of capacity across theatres and beds, and predicting the impact of productivity measures on capacity requirements. The model developed in this study is being used to provide quantitative support for accessing central capital funds, enabling discussion and evidence for the most efficient use of new resourcing. The model has been developed to offer a transferable solution for supporting both orthopaedic elective recovery, and with minor adaptations, recovery of other elective services.

**Contributors** AH drafted the original manuscript. AH, TM and MP developed the model. AH, TM, RW, MTR, EE, TJ, CP, AE, TK, MP, AB, MRW and AJ contributed to the conception and planning of this project, the acquisition of data, critically reviewed and revised the manuscript, and approved the final version. MTR and AJ supervised the project. MTR is the guarantor for the project. All authors agreed to be accountable for all aspects of the work.

**Funding** This study was funded by the Health Data Research (HDR) UK South West Better Care Partnership (#6.12). AH, TM and MP are supported by the National Institute for Health Research Applied Research Collaboration South West Peninsula. RW, EE, TJ and MTR's time was supported by the National Institute for Health Research Applied Research Collaboration West (NIHR ARC West). AJ, MRW, AB and CP were supported by the NIHR Biomedical Research Centre at University Hospitals Bristol and Weston NHS Foundation Trust and the University of Bristol.

**Disclaimer** The views expressed in this publication are those of the author(s) and not necessarily those of the National Institute for Health Research or the Department of Health and Social Care.

**Competing interests** None declared.

**Patient and public involvement** Patients and/or the public were involved in the design, or conduct, or reporting, or dissemination plans of this research. Refer to the Methods section for further details.

**Patient consent for publication** Not applicable.

**Ethics approval** We were provided with pseudonymised North Bristol Trust hospital admissions data under the NIHR ARC West Partnership Agreement. The project received ethical approval from the University of Bristol Faculty of Health Sciences ethical review board on 3 November 2020 (ref# 109024).

**Provenance and peer review** Not commissioned; externally peer reviewed.

**Data availability statement** Data may be obtained from a third party and are not publicly available. The data used in the study are collected by the North Bristol

NHS Trust (NBT) as part of their care and support. Sharing of anonymised data with the University of Bristol was underpinned by a data sharing agreement and solely covers the purposes of this study. Data requests can be made through the NBT. Code, input parameters and output data used for results in this paper are available on GitHub: https://github.com/AliHarp/HEP and archived using Zenodo https://zenodo.org/records/7951080. Code, results and testing can be viewed at https://aliharp.github.io/HEP/HEP_notebooks/01_intro.html. All are licensed using MIT permissive license MIT License | Choose a License. Not applicable.

**ORCID iDs**
Alison Harper http://orcid.org/0000-0001-5274-5037
Thomas Monks http://orcid.org/0000-0003-2631-4481
Rebecca Wilson http://orcid.org/0000-0003-4709-7260
Maria Theresa Redaniel http://orcid.org/0000-0002-0668-0874
Emily Eyles http://orcid.org/0000-0002-2695-7172
Tim Jones http://orcid.org/0000-0002-1199-8668
Chris Penfold http://orcid.org/0000-0001-8654-353X
Tim Keen http://orcid.org/0009-0005-6230-2455
Martin Pitt http://orcid.org/0000-0003-4026-8346
Ashley Blom http://orcid.org/0000-0002-9940-1095
Michael R Whitehouse http://orcid.org/0000-0003-2436-9024
Andrew Judge http://orcid.org/0000-0003-3015-0432

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
