## [Reviewer comments · BMJ Open]

ARTICLE DETAILS

TITLE (PROVISIONAL)	Development and application of simulation modelling for orthopaedic elective resource planning in England
AUTHORS	Harper, Alison; Monks, Thomas; Wilson, Rebecca; Redaniel, Maria Theresa; Eyles, Emily; Jones, Tim; Penfold, Chris; Elliott, Andrew; Keen, Tim; Pitt, Martin; Blom, Ashley; Whitehouse, Michael; Judge, Andrew

VERSION 1 – REVIEW

REVIEWER	Landa, Paolo Laval University, Operations and Decision Systems Department
REVIEW RETURNED	11-Aug-2023

GENERAL COMMENTS	The manuscript describes a DES model applied to an orthopaedic unit that perform surgeries to elective patients in an English hospital. The model is well explained and presents all the part quite well in the manuscript. I have few questions on the model and the assumptions: 1) it is not clear how works the priority. Are there classes? How you estimate them within patients?2) DES models work really well within this topic and application. However I don't have clear which is the novelty of this paper. Please can you state it better in the manuscript?3) a large contribution of the results depends also on the weekly schedule. Have you tried other weekly schedules in order to estimate the robustness of the model?4) in this study they do not include stochastic events (such as delays) in surgeries.5) have you included in the experiments the variation of the revision slots? How it impact on the results?6) the random sampling is based on normal distribution?7) In the indicator of the average rate of ward utilisation I would suggest also to include the standard deviation.
---

REVIEWER	Madanipour, S Royal Free London NHS Foundation Trust
REVIEW RETURNED	20-Sep-2023

GENERAL COMMENTS	The key findings of this developed model include: - The system is more sensitive to changes in procedure LOS than to changes in the proportion of patients with delayed discharge.- Bed utilisation remains high if one reduces delayed discharges but does not effect change in the LOS.- If LOS is reduced, then reduction in delayed discharge is much more effective in optimising bed utilisation.
--

	- This effect continues with the modelled additional weekend activity. - Where lengthsof-stay remain at baseline values, weekend theatre activity cannot be considered, as bed utilisation and resultan tlost slots are unacceptably high, even up to 70 beds. This model provides valuable information for workforce planners and hospital administrators. The model can therefore potentially prevent costly misadventure in the planning of weekend operating without first addressing baseline length of stay. Resources in trusts with above average length of stay can be prioritised in this area before considering expansion of capacity. It will be interesting to see how this prototypical model can be broadened: on the operative side to include parameters such as surgical complexity, availability of assistants, training cases, nursing staff seniority and availability of industry representatives and on the post operative side to include workforce availability and seniority of therapies and nursing staff to better understand how this is likely to impact LOS and therefore surgical throughput. The message of this paper is clear – administrators at all levels should be focussing on improving length of stay to better influence surgical throughput. What is likely to make this model more generalisable and effective at an individual unit level is the ability to input the parameters that impact that very length of stay. That being said, this is a valuable tool and an important step towards an algorithmic, generalisable approach to a national crisis that simply does not exist in comparable elective orthopaedic units in North America or Europe. This is therefore an important piece of work relating to a valuable modelling tool.
--	---

REVIEWER	Shetty, Vishvas HOME (Hiranandani Orthopaedic medical education , Orthopaedics
REVIEW RETURNED	29-Sep-2023

GENERAL COMMENTS	Overall, this looks like an excellent project. Just one question, in your model, are revision arthroplasty cases stratified by cause for revision i.e aseptic loosening, infection, instability, trauma etc ? The average length of stay, operative times and same day cancellation rates would vary between the groups and might affect the statistical modelling. Thank you.
---

VERSION 1 – AUTHOR RESPONSE

Reviewer: 1 Dr. Paolo Landa, Laval University

The manuscript describes a DES model applied to an orthopaedic unit that perform surgeries to elective patients in an English hospital. The model is well explained and presents all the part quite well in the manuscript.

Many thanks for your kind comments.

1) it is not clear how works the priority. Are there classes? How you estimate them within patients?

There is a FIFO priority system in the model; please see Section 2.4 of STRESS-DES Reporting Guidelines, Supplementary Material 1.

2) DES models work really well within this topic and application. However I don't have clear which is the novelty of this paper. Please can you state it better in the manuscript?

Thank you for highlighting this lack of clarity.

We have added the following (highlighted) text to DISCUSSION:

Previous modelling and simulation work has focused on resource sharing for elective and non-elective joint replacement [17] and detailed studies of individual orthopaedic services [18, 23]. Although the need for reusable models for orthopaedic wait list planning is recognised [24], **this is the first** free and open-source, **generalisable DES** model to support resourcing and efficiency of elective services **that is available for use in this area** [26]. Our model provides quantitative outputs estimating the effects on surgical throughput (per procedure); daily bed utilisation of changes to bed and theatre capacity, theatre scheduling, patient lengths-of-stay and discharge delays; and lost theatre slots, representing system pressure on beds. The model **is designed** for use by health services planners and clinicians, and is available as a free web-app to address usability and accessibility of results [27]. It is being used to evidence service configurations for the business case in NBT, and is more widely applicable.

3) a large contribution of the results depends also on the weekly schedule. Have you tried other weekly schedules in order to estimate the robustness of the model?

Many thanks for this comment – yes, a significant amount of sensitivity analysis and validation was performed using a variety of parameter combinations. These are available in the links provided in the paper, and can be viewed in our online notebook (linked in the GitHub repository):

https://aliharp.github.io/HEP/HEP_notebooks/01_intro.html

We have added the following sentence to the end of the results section:

Sensitivity analysis and testing of parameters has been undertaken and is available to view in GitHub, Zenodo, and our online notebook (see Data Availability Statement).

We have also linked the online notebook in the Data Availability Statement to make it easier to view these results.

4) in this study they do not include stochastic events (such as delays) in surgeries.

This is true – the DES and surgical schedule run in days. The model assumes that if a patient is scheduled for surgery it will be undertaken during that day, unless there is no bed available. However delayed lengths-of-stay are determined using stochastic routing. We have added a line in to clarify this:

“Using stochastic routing, a length-of-stay is sampled for delayed patients.”

5) have you included in the experiments the variation of the revision slots? How it impact on the results?

In this paper, these results are not presented, however they are available in the GitHub repository and can be viewed in our online notebook.

6) the random sampling is based on normal distribution?

Using 'fitter' package in Python, *lognormal* distributions were fitted for all lengths-of-stay. The code for this is available in the linked repository and online notebook. To clarify this, we have added an additional sentence in the Data and Setting section:

“Lognormal distributions were used for sampling lengths-of-stay in all cases.”

Please note, this is also mentioned in the section 'Orthopaedic surgical pathway': “The DES samples a length-of-stay for each simulated patient from a lognormal distribution with parameters calculated from the previously fitted distributions by procedure type.”

7) In the indicator of the average rate of ward utilisation I would suggest also to include the standard deviation.

Thank you, we agree that variability in utilisation is an important output. In this paper, the indicative results we presented show mean outputs as the figure contains a lot of information (72 scenarios) and is intended to demonstrate the applicability of the model. Within the web app, which is designed for direct use by healthcare users, results for bed utilisation include variability in output using boxplots.

We have added the following line to the end of the results section:

Within the web application, results for bed utilisation include variability in output using boxplots.

Reviewer: 2

Dr. S Madanipour, Royal Free London NHS Foundation Trust

This model provides valuable information for workforce planners and hospital administrators. The model can therefore potentially prevent costly misadventure in the planning of weekend operating without first addressing baseline length of stay. Resources in trusts with above average length of stay can be prioritised in this area before considering expansion of capacity. It will be interesting to see how this prototypical model can be broadened: on the operative side to include parameters such as surgical complexity, availability of assistants, training cases, nursing staff seniority and availability of industry representatives and on the post operative side to include workforce availability and seniority of therapies and nursing staff to better understand how this is likely to impact LOS and therefore surgical throughput.

Many thanks – we agree that there is a lot more work that can be undertaken in this area, including workforce availability.

The message of this paper is clear – administrators at all levels should be focussing on improving length of stay to better influence surgical throughput. What is likely to make this model more generalisable and effective at an individual unit level is the ability to input the parameters that impact that very length of stay. That being said, this is a valuable tool and an important step towards an algorithmic, generalisable approach to a national crisis that simply does not exist in comparable elective orthopaedic units in North America or Europe. This is therefore an important piece of work relating to a valuable modelling tool.

We appreciate your kind comments and recognition of the potential value of this work.

Reviewer: 3

Dr. Vishvas Shetty, HOME (Hiranandani Orthopaedic medical education)

Overall, this looks like an excellent project. Just one question, in your model, are revision arthroplasty cases stratified by cause for revision i.e aseptic loosening, infection, instability, trauma etc ? The average length of stay, operative times and same day cancellation rates would vary between the groups and might affect the statistical modelling. Thank you.

Very many thanks for this important comment. In this model we chose not to further stratify surgical types, although the underlying data would support this (and it is possible for researchers/healthcare analysts who choose to build on this work to adapt the model accordingly using the available code).

We chose not to further stratify as the primary focus of the model is the system as a whole and not on individual groups, so the non-stratified model is sufficient to meet the goals of the model in this application.

Note that the lengths-of-stay for each surgical type are represented by probability distributions (log normal) which capture the variability in lengths-of-stay seen across surgical types (e.g. revision hip), including the few very long stays (i.e. we do not model using means). We chose not to truncate the underlying data when fitting distributions to ensure all variability is captured – so when running the model, there is a low probability that patients may sample long lengths-of-stay, representing those patients whose surgery and recovery are more complex.

Variation in operative times is a factor for consideration by surgeons using the model (and defining the baseline schedule).

Same day cancellation rates for patient reasons are not considered in this model, except as a proportion of 'lost slots'. We were not able to access data on same day cancellations per surgical type or surgical cause. A risk of adding this additional complexity is that within the model it will show more beds to be available, while in practice, more efforts are being made to reduce same day cancellations for patient reasons (such as enhanced pre-surgical activities).

I hope this clarifies our modelling decisions, and thank you very much for your interest.